# Detection of Electromagnetic Seismic Precursors from Swarm Data by Enhanced Martingale Analytics

**DOI:** 10.3390/s24113654

**Published:** 2024-06-05

**Authors:** Shane Harrigan, Yaxin Bi, Mingjun Huang, Christopher O’Neill, Wei Zhai, Jianbao Sun, Xuemin Zhang

**Affiliations:** 1School of Computing, Engineering, and Intelligent Systems, Ulster University, Derry-Londonderry BT48 7JL, UK; sp.harrigan@ulster.ac.uk; 2School of Computing, Ulster University, Belfast BT15 1AP, UK; oneill-c177@ulster.ac.uk; 3School of the Built Environment, Ulster University, Belfast BT15 1AP, UK; m.huang@ulster.ac.uk; 4Lanzhou Institute of Geotechnique and Earthquake, China Earthquake Administration, Lanzhou 730000, China; zhaiw@gsdzj.gov.cn; 5Institute of Geology, China Earthquake Administration, Beijing 100017, China; sunjianbao@gmail.com; 6Institute of Earthquake Forecasting, China Earthquake Administration, Beijing 100036, China

**Keywords:** anomaly detection, Martingale theory, electromagnetic seismic precursors, Swarm satellites, earthquake

## Abstract

The detection of seismic activity precursors as part of an alarm system will provide opportunities for minimization of the social and economic impact caused by earthquakes. It has long been envisaged, and a growing body of empirical evidence suggests that the Earth’s electromagnetic field could contain precursors to seismic events. The ability to capture and monitor electromagnetic field activity has increased in the past years as more sensors and methodologies emerge. Missions such as Swarm have enabled researchers to access near-continuous observations of electromagnetic activity at second intervals, allowing for more detailed studies on weather and earthquakes. In this paper, we present an approach designed to detect anomalies in electromagnetic field data from Swarm satellites. This works towards developing a continuous and effective monitoring system of seismic activities based on SWARM measurements. We develop an enhanced form of a probabilistic model based on the Martingale theories that allow for testing the null hypothesis to indicate abnormal changes in electromagnetic field activity. We evaluate this enhanced approach in two experiments. Firstly, we perform a quantitative comparison on well-understood and popular benchmark datasets alongside the conventional approach. We find that the enhanced version produces more accurate anomaly detection overall. Secondly, we use three case studies of seismic activity (namely, earthquakes in Mexico, Greece, and Croatia) to assess our approach and the results show that our method can detect anomalous phenomena in the electromagnetic data.

## 1. Introduction

The history of seismic event detection spans millennia, with early records tracing back to ancient China around 2000 years ago [1]. However, significant advancements in seismological studies emerged predominantly in the mid 20th century, driven largely by the interests of superpowers engaged in nuclear weapon testing [2]. Over recent years, technological progress has led to the deployment of powerful sensors on both ground-based stations and satellites orbiting the Earth [3,4,5]. This has contributed to a growing body of evidence suggesting a connection between seismic activity and electromagnetic phenomena. Notably, the Rikitake Law, discovered by Japanese physicist Rikitake, demonstrates a linear relationship between the logarithm of electromagnetic precursor time and earthquake magnitude [6].

This paper focuses on detecting abnormalities in electromagnetic field behavior measured by satellites, with a specific interest in the Swarm constellation operated by the European Space Agency (ESA) since 2013 [7]. Swarm, comprising three identical satellites equipped with high-precision electromagnetic field sensors, was originally designed to study ionosphere behavior for space physics research. However, it has also been utilized in research examining electromagnetic field variations associated with seismic events.

The Swarm satellite constellation comprises three satellites: Alpha, Beta, and Charlie. They are identical, differing only in altitude and orbit behavior. Alpha and Charlie orbit roughly 450 km high, with a 100-km separation, while Beta orbits at around 510 km, nearly perpendicular to Alpha and Charlie [7]. Originally, Beta was intended to orbit alongside Alpha, but due to issues with Charlie, their altitudes were swapped, although this did not affect the constellation functionality. Each satellite records electromagnetic activity every second, along with corresponding geospatial coordinates (latitude and longitude). Consequently, each satellite produces 86,400 observations per day, totaling 259,200 for all satellites. This amounts to approximately 756,280,000 available records detailing electromagnetic field (EMF) behavior. From this extensive dataset, we focus on a subset within specific geospatial bounds for our experiments.

This paper introduces an improved abnormal change detection algorithm based on Martingale probability theory [8,9]. Unlike the original approach, which required a predefined threshold, our method does not, reducing errors introduced during data aggregation. We conduct two experiments: a quantitative analysis to evaluate our method performance against benchmark data and a qualitative study using real Swarm data to investigate three seismic events—the 2017 Mexican earthquake, the 2020 Greek earthquake, and the 2020 Croatian earthquake. Our results suggest that our approach can detect abnormal EMF patterns preceding seismic events, sometimes months in advance. We believe these findings provide a foundation for developing abnormal change detection algorithms for space-based EMF time-series data.

Large-magnitude earthquakes, while rare, have significant societal impacts in terms of loss of life and economic damage. They result from cumulative strain over time, culminating in lithospheric rupture and bedrock displacement [10]. Studies have shown seismic anomalies occurring before earthquakes, termed precursors or abnormal signals, suggesting an energy exchange between the lithosphere and the ionosphere—the ionized upper atmosphere. Ground-based and space-borne measurements offer insights into precursor signals. Space-borne measurements, like those from CHAMP, DEMETER, and Swarm satellites, provide valuable data for studying ionospheric activity and its relation to seismic events. Swarm data, in particular, have shown promise in detecting seismic event precursors through the analysis of EM variations [11,12,13]. This paper presents an enhanced Martingale algorithm for detecting abnormal changes in EMF data. We focus on probabilistic approaches for anomaly detection, as they do not require predefined thresholds, which is advantageous given the difficulty in defining anomalies within the EMF domain. Building on previous work, we propose a novel method utilizing Martingale probability theory, offering a robust probabilistic model without the need for predefined thresholds [14,15].

## 2. Electromagnetic Anomaly Detection via Martingale Systems

Martingale probability theory models a fair system where, in gambling terms, only one successful bet is needed to recover from previous losses. A sequence of random variables $V_{i}$, where *i* ranges from some starting point to infinity, is considered a Martingale with respect to another sequence of random variables $Z_{i}$ if certain conditions are met. Specifically, for all i≥0, $V_{i}$ must be a measurable function of $Z_{0},\ldots ,Z_{i}$, the expected absolute value of $V_{i}$ must be finite, and the conditional expectation of $V_{n+1}$ given $Z_{1},\ldots ,Z_{n}$ is equal to $V_{n}$, indicating that *V* is expressible in terms of *z* with finite expectation. In simpler terms, a Martingale is a sequence where each new value is, on average, equal to the previous value.

In our problem domain, we are interested in exchangeability testing using Martingale theory [8,16]. Exchangeability refers to a property of the joint distribution of $Z_{i}$, where for any subset of $Z_{i}$ the joint distribution remains the same under permutation. In other words, the order of the sequence does not affect the probability distribution. To assess exchangeability, we utilize a strangeness measure, as described in [9], which evaluates how unusual a data point is compared to others in its set. Since anomaly detection is typically unsupervised, we employ a cluster-based strangeness measure incorporating distance metrics.

Seismic events, such as earthquakes, are often preceded by various precursor phenomena, including changes in electromagnetic fields. These electromagnetic anomalies arise due to the complex interactions between tectonic plates, stress accumulation, and the release of energy along fault lines. As seismic activity generates stress within the Earth’s crust, it can lead to the fracturing of rocks, which in turn produces electrical charges and perturbations in the surrounding electromagnetic field.

Detecting electromagnetic anomalies as precursors to seismic events offers valuable insights into the underlying processes leading to earthquakes. However, it is crucial to address several key questions regarding this approach. Firstly, can electromagnetic anomalies reliably serve as indicators of impending seismic activity? Research suggests that while electromagnetic precursors show promise, their predictive value may vary depending on factors such as the proximity to the epicenter and the magnitude of the impending earthquake.

Furthermore, the use of satellite-based electromagnetic monitoring systems raises questions about detection distances and false positives. Satellites equipped with electromagnetic sensors can capture data over large geographic areas, allowing for wide-scale monitoring. However, it is essential to ensure that detected anomalies are genuinely related to seismic activity and not incidental variations in the electromagnetic field.

To address these concerns, our approach incorporates rigorous statistical methods, such as Martingale theory, to analyze electromagnetic data and identify anomalies indicative of seismic precursors. By employing cluster-based strangeness measures and considering spatial and temporal relationships, we aim to distinguish genuine anomalies from background noise and incidental fluctuations. This rigorous approach enhances the reliability of our anomaly detection system and contributes to the understanding of electromagnetic precursors to seismic events.

For an unlabeled training set $T=x_{1},\ldots ,x_{n}$, such as the EMF data from Swarm satellites, the strangeness of $x_{i}$ is defined as $s\left(T,x_{i}\right)=\left|\right|x_{i}-c\left|\right|$, where *c* is the centroid of *T* and ||·|| denotes a distance metric. This strangeness measure represents the distance of a data point from the centroid of its cluster, serving as a measure of its abnormality.

Using $s(T,x_{i})$, we can compute the $\hat{p}$-value for each data point using the following formula:(1)$$\hat{p}=\frac{\#\left\{j:s_{j}>s_{i}\right\}+\theta_{i}\#\left\{j:s_{j}=s_{i}\right\}}{i}$$
Here, $\theta_{i}$ is a random number from the interval [0, 1] (e.g., 0.5) at instance i=1,2,…,n, and $s_{j}$ represents the strangeness measure for j=1,…,i. The symbol “#” denotes the cardinality of the dataset, counting the number of *j* for which $s_{j}>s_{i}$.

Using these $\hat{p}$-values, we construct a randomized power Martingale indexed at ϵ∈[0,1]:(2)$$M_{n}^{\epsilon }=\prod_{i=1}^{n}\left(\epsilon p_{i}^{\epsilon -1}\right)$$
Alternatively, as noted in [9], we observe that
(3)$$M_{n}^{\epsilon }=\epsilon p_{i}^{\epsilon -1}M_{i-1}^{\epsilon }$$

This formulation prevents redundant computations for consecutive $\hat{p}$-values. We test exchangeability by comparing the resulting Martingale $M_{n}^{\epsilon }$ to a predefined threshold λ, assuming a null hypothesis $H_{0}$: “no change in the data”, against the alternative $H_{1}$: “change in the data”:(4)$$0<M_{n}^{\epsilon }<\lambda$$
If $M_{n}^{\epsilon }>\lambda$, $H_{0}$ is rejected.

In our novel approach, we enhance this process by first implementing the Gaussian Moving Average Martingale (GAMAM). For the *n*-th set, GAMAM produces a Gaussian kernel $\psi_{i}$:(5)$$\psi_{i}=\frac{1}{\sigma \sqrt{2\pi }}e^{-\frac{1}{2}{\left(\frac{M_{n}^{\epsilon }}{\sigma }\right)}^{2}}$$
where, σ is the standard deviation of the distribution, assumed to have a mean of zero. We then replace the Martingale point $M_{n}^{\epsilon }$ with a smoothed value $G_{k}$ using the Gaussian kernel:(6)$$G_{k}=\sum_{i=0}^{k}\psi_{i}M_{k-i}^{\epsilon }$$

This smoothing operation allows us to compute λ by calculating the *z*-score of each $G_{k}$ and determining the mean absolute deviation (MAD) of that point:(7)$$J_{z}=\frac{M_{m}-\mu }{\sigma }$$
where, z=1,…,n and m=1,…,n, respectively. If a point exceeds the MAD, it signifies a change or anomaly in the data. Consequently, we express $H_{0}$ as satisfied as long as $0<J_{z}<\lambda$, where λ=MAD. This approach, utilizing the *z*-score, eliminates the need for a predefined λ, as it is calculated dynamically based on the characteristics of the evolving dataset.

Algorithm 1 elucidates our proposed methodology, while Figure 1 visually depicts the algorithm workflow through a detailed flowchart. This graphical representation offers a clear and concise illustration of the sequential steps involved in our approach, enhancing the understanding of our proposed method.
**Algorithm 1** *z*-Martingale Algorithm**Require:** T=⌀**Ensure:** $M_{0}=1$, i=1 y←1 X←x N←n **loop**   **if** T=⌀ **then**▹ operate while $x_{i}$ is new     $s(T,x_{i})=0$   **else**     Compute strangeness $s(T,x_{i})$   **end if**   Compute ${\hat{p}}_{i}$ using (Equation 1)   Compute $M_{i}^{\epsilon }$ using (Equation 3)   Compute $J_{z}$ for n=1,…,i using (Equation 7)   λ←MAD   **if** $M_{z}>\lambda$ **then**▹ If $M_{z}$ exceeds λ then trigger alarm     Return **true**   **end if**   i←i+1 **end loop**

## 3. Experiments

We conduct two experiments to evaluate the performance of our novel approach, each discussed sequentially.

The first experiment is quantitative and aims to provide accuracy and *r*-rank metrics. Its primary objective is to facilitate comparative analysis between our novel approach and the original changepoint detection framework upon which we build. For this experiment, we utilize five widely recognized benchmark datasets containing anomalies [17]. We assess our algorithm’s ability to detect these anomalies without prior knowledge. Given the probabilistic nature of our linear model, we anticipate that our algorithm should compete directly with the original approach as it traverses the data over time. The second experiment is qualitative and revolves around three case studies of real-world large magnitude earthquakes: the 2017 Mexican earthquake, the 2020 Greek earthquake, and the 2020 Croatian earthquake. For each experiment, we outline the parameters used when running our algorithm over the data. Unless otherwise specified, these parameters remain constant throughout the experiment. Additionally, we provide discussions on the results obtained at the end of each experiment.

### Quantitative Experiment

To analyze the developed approach, we devised two experiments. The first experiment is geared towards obtaining quantitative metrics such as accuracy, precision, and *r*-rank for both the original Martingale framework [9] and the novel approach presented in this work. Standard statistical measures are employed to compute the evaluation metrics of accuracy and precision.

Specifically, we define the real positives (*P*) and real negatives (*N*) of the data, creating two groups. Following standard practice, data are split into true positives (TP), true negatives (TN), false positives (FP), and false negatives (FN); these definitions form the basis of precision and accuracy calculations. Precision reflects the number of correct identifications within the positive (TP and FP) categories and is defined as
(8)$$\mathrm{Precision}=\frac{\mathrm{TP}}{\mathrm{TP}+\mathrm{FP}}$$

Accuracy reflects the correctness of the overall results, including both positive and negative outcomes, and is defined as
(9)$$\mathrm{Accuracy}=\frac{\mathrm{TP}+\mathrm{TN}}{\mathrm{TP}+\mathrm{FP}+\mathrm{TN}+\mathrm{FN}}$$

Accuracy measures the overall correctness of the results, considering both positive and negative outcomes. It quantifies the proportion of correctly classified instances among all instances, encompassing true positives (correctly identified anomalies) and true negatives (correctly identified non-anomalies), over the total number of instances. Precision, on the other hand, focuses specifically on the accuracy of the positive results. It quantifies the proportion of correctly identified anomalies (true positives) among all instances classified as anomalies, including both true positives and false positives (instances wrongly identified as anomalies).

The *r*-rank, also known as the R-metric, is a metric [18] gaining popularity in contexts similar to the problem areas addressed in this paper. It involves assessing the accuracy of individual methods using a predefined length. The *r*-rank takes into account the localization of anomalies by ranking them and serves as a measure of accuracy by comparing expected anomalies with detected anomalies in terms of importance, typically using a ranked nearest neighbor approach. Essentially, it evaluates how accurately the detected anomalies are ranked compared to the true known anomalies, reflecting the method’s performance.

Our first experiment is aimed at assessing the efficacy of our algorithm in detecting anomalies following a common experimental design in this scenario space [19]. We utilized a prominent electrocardiogram (ECG) database known as the “BIDMC Congestive Heart Failure Database” (CHFDB) [20,21,22,23,24], which is widely recognized for its well-documented anomalies (via a trained autoencoder) and annotated time and position data. This database comprises 15 recordings obtained from a diverse cohort of participants, including 11 men and 4 women, who suffer from severe congestive heart failure. Each recording spans approximately 20 h, with a sampling rate of 250 Hz and a 12-bit resolution. From the CHFDB, we utilize four datasets with known and understood anomalies [19]. Figure 2 shows a sample trace of the ECG used in this experiment.

The CHFDB is a highly regarded resource in cardiology research, providing a comprehensive collection of ECG recordings from patients with congestive heart failure. These recordings offer valuable insights into the electrical activity of the heart during various stages of cardiac dysfunction, facilitating the development and validation of algorithms for detecting abnormal cardiac rhythms and anomalies. The database includes detailed annotations and metadata, enabling precise analysis and interpretation of the recorded ECG signals. Extensive prior studies have demonstrated that ECG data share several similarities with EMF data [25,26]. Additionally we include another EMF dataset, “Data4”, which was collaboratively obtained [2].

Table 1 and Table 2 showcase the outcomes of employing two distinct anomaly detection methodologies: the original Martingale and the *Z*-score enhanced Martingale, on multiple datasets. These methodologies were evaluated based on three key performance metrics: accuracy, precision, and R-rank. The R-rank, also known as the Relative Rank, is a metric used to evaluate the performance of a classification model. It represents the relative position of the true positive rate (sensitivity) against the false positive rate (1 − specificity) on a receiver operating characteristic (ROC) curve. To calculate R-rank, we take the area under the ROC curve and divide by the total area of the ROC chart.

Regarding the original Martingale results (Table 1), diverse performance levels are evident across different datasets. For instance, dataset Chfdb_chf01_275_f1 attained an accuracy of 91% and a precision of 79%, signifying a relatively high accuracy in identifying anomalies, albeit with some false positives. However, the R-rank for this dataset is notably low at 26%, indicating that the detected anomalies may not consistently rank highly in importance or severity. Conversely, dataset Chfdb_chf13_45590_f2 exhibited higher precision (75%) and R-rank (28%) with a slightly improved accuracy of 94%, indicating better performance in correctly identifying anomalies with higher importance.

Transitioning to the results of the *Z*-score enhanced Martingale (Table 2), a similar trend in performance metrics is observed with some variations. For instance, in dataset Chfdb_chf01_275_f1, the *Z*-score enhanced Martingale achieved the same accuracy (91%) as the original Martingale but with a slightly lower precision (75%). However, the R-rank notably increased to 62%, indicating that the detected anomalies are more likely to be of higher importance or severity compared to the original Martingale results. Similarly, in dataset Chfdb_chf13_45590_f2, the *Z*-score enhanced Martingale demonstrated perfect precision (100%) and a significantly higher R-rank (55%) compared to the original Martingale results, suggesting a more effective prioritization of detected anomalies.

Overall, these results underscore the potential advantages of employing the *Z*-score enhanced Martingale approach, particularly in enhancing the ranking of detected anomalies based on their importance or severity. Nonetheless, further analysis and validation are essential to comprehensively understand the implications of these performance disparities and to evaluate the robustness of the *Z*-score enhanced Martingale approach across various datasets and contexts.

## 4. Quantitative Experiment

As discussed earlier, Swarm satellites generate a significant amount of data at second intervals, accumulating a large dataset since project inception and the initial report observations. For our study, we have chosen to focus on a limited time frame—specifically, one year prior to the seismic event—including the day of the seismic event itself and 3-month aftershocks. However, in future studies, we may extend our observations further into the past. Based on the success of our algorithm in the first experiment, we can assert that our algorithm is capable of detecting anomalies with a high degree of accuracy and correctness, with our hypothesis being that our algorithmic system is transferable into this problem domain. Building upon this, we examine the EMF activity in relation to time preceding the seismic event, such as three months, six months, and twelve months, as well as three months post-seismic event. To refine our analysis, we narrow down the dataset by imposing a geospatial constraint, meaning only data within predefined latitude and longitude grids are considered for forming the time-series analyzed by our algorithm. Specifically, we utilize two grids: a 1000km×800km grid and a 500km×300km grid, respectively.

The Swarm satellite constellation, comprising satellites Alpha, Beta, and Charlie [26], operates with distinct orbits: A and C travel at an altitude of approximately 450 km in parallel, while B orbits at around a 550 km altitude, perpendicular to Alpha and Charlie. These satellites record Vector Field Magnetometer (VFM) readings, capturing three-axis bounded values denoted as $b_{x}^{2}$, $b_{y}^{2}$, and $b_{z}^{2}$. To simplify analysis, we reduce the data dimensionality in a 1-dimensional intensity $|\vec{B}|$ value defined as
(10)$$|\vec{B}|=\sqrt{b_{x}^{2}+b_{y}^{2}+b_{z}^{2}}$$

This transformation enables a more streamlined approach to analyzing the magnetic field data, facilitating subsequent interpretation for various applications, such as seismic event detection. Utilizing (Equation 10), we define the magnitude of the intensity values across a single day of samples *N* as
(11)$$\tilde{B}=\sum_{i=1}^{N}{\left|\vec{B}\right|}_{i}$$

Taking (Equation 11), we can further define the average intensity value over multiple days (complying with our algorithmic *T*) as
(12)$$T=\left\{\frac{1}{N_{j}}{\tilde{B}}_{j}|j=1,2,\ldots ,J\right\}$$

Our final time-series, per satellite, comprises data points representing the mean average across the orbits of a given day. It is important to note that non-orbit days are excluded from the final time-series. This ensures that our analysis focuses on data captured during satellite orbits, providing a more accurate representation of the electromagnetic field activity.

Furthermore, by constraining our analysis to specific geospatial grids, we aim to enhance the precision of our findings by focusing solely on regions of interest where seismic activity is most likely to occur. This approach allows us to zoom in on areas with higher seismic risk, potentially increasing the sensitivity of our anomaly detection algorithm to relevant electromagnetic field fluctuations preceding seismic events. Additionally, the aggregation of data points to form daily mean averages helps smooth out noise and fluctuations, providing a clearer signal for anomaly detection. This preprocessing step is essential for ensuring the robustness and reliability of our analysis, particularly when dealing with high-frequency data generated by Swarm satellites.

By incorporating both temporal and spatial dimensions into our analysis, we can better understand the relationship between electromagnetic field activity and seismic events. This multidimensional approach enables us to uncover potential patterns and correlations that may have previously gone unnoticed, ultimately contributing to our understanding of seismic precursors and improving our ability to forecast and mitigate earthquake risks.

Our case studies focus on the earthquakes that occurred in Mexico (19 September 2017—https://earthquake.usgs.gov/earthquakes/eventpage/us2000ar20/executive (accessed on 10 March 2021)), Greece (30 October 2020—https://earthquake.usgs.gov/earthquakes/eventpage/us7000c7y0/executive (accessed on 10 March 2021)), and Croatia (29 December 2020—https://earthquake.usgs.gov/earthquakes/eventpage/us6000d3zh/executive (accessed on 10 March 2021)).

The earthquake in Mexico occurred on 19 September 2017, with its epicenter—latitude $18.550^{\circ }\;N$ and longitude $98.489^{\circ }\;W$—located near the town of Raboso in the state of Puebla, approximately 120 km southeast of Mexico City. The earthquake, with a depth of 48 km, had a magnitude of 7.1 and caused widespread devastation, resulting in hundreds of fatalities and significant damage to buildings and infrastructure in Mexico City and the surrounding areas.

The earthquake in Greece struck on 30 October 2020, with its epicenter—latitude $37.897^{\circ }\;N$ and longitude $26.784^{\circ }\;E$—situated near the island of Samos in the eastern Aegean Sea. The earthquake, with a depth of 21 km, had a magnitude of 7.0 and was felt across a wide area, including parts of Greece and Turkey. It caused buildings to collapse, resulting in casualties and injuries, particularly in the city of Izmir, Turkey, and on the island of Samos.

The earthquake in Croatia occurred on 29 December 2020, with its epicenter—latitude $45.424^{\circ }\;N$ and longitude $16.257^{\circ }\;E$—located near the town of Petrinja in central Croatia. The earthquake, with a depth of 10 km, had a magnitude of 6.4 and caused significant damage to buildings and infrastructure in Petrinja and surrounding areas. It resulted in several fatalities and numerous injuries, as well as widespread displacement of residents.

We present the outcomes of the case studies in a methodical, chronological order, meticulously sorted by ascending grid size. Each geospatial boundary grid underwent a thorough analysis, yielding three distinctive plots showcasing the original EMF data. These plots include (1) the unaltered raw data, providing a baseline perspective; (2) the data augmented with anomalies identified by the Martingale framework overlaid, highlighting potential disruptions or irregularities; and (3) the anomalies identified by our innovative framework overlaid, offering further insights into nuanced variations.

Within each plot, a comprehensive examination of observable EMF characteristics is conducted. We meticulously document instances of positive and negative spikes, indicating notable increases and decreases in intensity values, respectively. These spikes are scrutinized to discern their significance within the broader context of the plot, considering factors such as magnitude and duration. Additionally, the temporal proximity of each spike to the seismic event is meticulously annotated, employing vertical reference lines to denote specific time intervals (e.g., 3 months prior and 3 months post-event). This in-depth annotation facilitates a detailed temporal analysis, allowing for the identification of potential precursory EMF patterns preceding seismic events and their evolution over time.

The same method is applied in each instance. Using Equation (Equation 10), we form a sequence of time-series data, wherein each data point is the mean of $|\vec{B}|$ values occurring over a day’s orbit(s). From this, we form two time-series sequences by varying the geospatial bound to 500×300km and 1000×800km. These are our ground truth data. The grid size will contribute significantly to the calculated $|\vec{B}|$ values because of its size as it changes the number of data points available over each day. We then examine the results of the conventional Martingale approach and compare them with the results of the proposed *z*-Martingale approach to determine which of the two methods is more accurate.

Within our case studies, the dataset *T* encompasses values spanning from one year preceding the seismic event to three months following it. This extensive temporal scope allows us to capture the dynamics of electromagnetic field (EMF) activity leading up to and after the seismic event, providing a comprehensive understanding of the pre-seismic and post-seismic EMF behavior. By including this wide range of temporal data, we aim to analyze and identify any discernible patterns, anomalies, or fluctuations in EMF intensity that may serve as potential precursors or indicators of seismic activity. This approach enables us to explore not only the immediate events surrounding the seismic occurrence but also the broader context and temporal evolution of EMF phenomena, thereby enhancing our ability to detect and interpret anomalies in the EMF data with greater depth and accuracy.

Our overarching objective encompasses not only the detection of anomalies within the data but also the assessment of their significance. As we progress through each case study, we meticulously evaluate the anomalies detected by our algorithms, aiming to discern their relevance and potential implications.

By scrutinizing the detected anomalies, we seek to identify those that exhibit notable characteristics or patterns, indicating a departure from expected behavior. Our assessment includes considerations such as the magnitude, frequency, and temporal distribution of anomalies, as well as their spatial correlation with known seismic events or geological features.

Furthermore, we delve into the context surrounding each anomaly, exploring factors such as its proximity to seismic activity, its alignment with historical events, and its correlation with environmental or anthropogenic factors. Through this comprehensive analysis, we endeavor to distinguish between anomalies of genuine interest and those that may arise from noise or artifacts in the data. Ultimately, our goal is to not only detect anomalies but also to interpret their significance within the broader context of seismic monitoring and environmental surveillance. By identifying and characterizing anomalies of particular interest, we aim to advance our understanding of electromagnetic phenomena and contribute valuable insights to the field of geoscience.

For brevity, we include the full results now for our case studies in Table 3. Table 3 provides a summary of the anomaly detection numbers for both the Martingale and *Z*-Martingale methods across the various case studies and grid sizes. In the Mexico case study (Section 4.1), using a grid size of 500×300 km, the Martingale method detected three anomalies while the *Z*-Martingale method detected four anomalies. When the grid size was increased to 1000×800 km, the Martingale method detected five anomalies and the *Z*-Martingale method detected five anomalies as well.

In the Greece case study (Section 4.2), with a grid size of 500×300 km, the Martingale method detected 21 anomalies while the *Z*-Martingale method detected 10 anomalies. Similarly, with a grid size of 1000×800 km, the methods detected twenty-one and nine anomalies, respectively.

For the Croatia case study (Section 4.3), using a grid size of 500×300 km, the Martingale method detected 17 anomalies, and the *Z*-Martingale method detected 6 anomalies. With a grid size of 1000×800 km, the Martingale method detected 20 anomalies, while the *Z*-Martingale method detected 10 anomalies.

### 4.1. Mexico 2017

The seismic activity of Mexico is deeply rooted in its geographical composition, being located atop several intersecting tectonic plates. The convergence of the Cocos Plate and North American Plate along the Pacific Coast, alongside activity along the edges of the Rivera and Caribbean plates, generates approximately 40 earthquakes daily in the country. The susceptibility of Mexico City to seismic events is exacerbated by its foundation on a dry lake bed characterized by soft soil comprising sand and clay. This geological makeup amplifies the destructive impact of major earthquakes, as loose sediments near the surface slow down shock waves, increasing both their amplitude and duration.

On 19 September 2017, at approximately 6:15 p.m. local time, the Ayutla region in Mexico, situated southeast of Mexico City, experienced a seismic event of considerable magnitude. The impact of the earthquake was profound, resulting in widespread damage categorized with a severity rating of VII (severe) intensity. The aftermath of the earthquake left a lasting imprint on the region, with buildings damaged, infrastructure disrupted, and communities in disarray. To visually contextualize the epicenter of the Mexico 2017 earthquake, a detailed plot of the 500×300km area of interest is provided in Figure 3.

#### 4.1.1. 500×300 km

Figure 4 illustrates the EMF intensity activity leading up to and following the seismic event. As described earlier, each data point represents the mean value aggregated across daily recordings within a 500×300 km grid, calculated according to Equation (Equation 11).

In Figure 5a, we illustrate the outcomes obtained from applying the original Martingale framework to the EMF data depicted in Figure 4. Despite the presence of approximately 15 significant anomalies in the ground truth data represented in Figure 4, the Martingale framework identifies only one anomaly in the EMF signals leading up to the seismic event. Remarkably, this sole anomaly is detected within the EMF data originating from the Alpha satellite. Conversely, subsequent to the seismic event, the Martingale framework identifies two anomalies. Intriguingly, both of these anomalies are observed within the EMF data derived from the Beta satellite and coincide temporally with the anticipated aftershock spikes, as depicted in Figure 4.

Figure 5b depicts the anomalies identified by our novel approach applied to the data visualized in Figure 4. Notably, there is a discernible escalation in the number of detected anomalies as the seismic event date approaches. Around nine months preceding the event, aberrant patterns in the time-series of Charlie are detected, followed by anomalies in the Beta time-series just under six months before. Subsequently, there is a surge in anomalies detected in the Charlie time-series just under three months before the event, accompanied by several anomalies in the Alpha time-series. Post the seismic event, anomalies are observed in the Beta time-series.

In contrast to the original Martingale framework illustrated in Figure 5a, our approach demonstrates a higher capability in detecting abnormalities across the time-series of all three satellites. Intriguingly, some detected spikes originate from regions that exhibit no apparent spikes or other anomalies in Figure 5. This observation suggests the presence of underlying data behavior in those regions that contravene the null hypothesis, serving as the triggering mechanism for an anomaly label.

#### 4.1.2. 1000×800 km

Figure 6 presents the EMF intensity activity over a larger area of 1000×800 km, both preceding and following the seismic event. Each data point in the graph represents the mean intensity values calculated using Equation (Equation 10) for a single day. A distinct shift in EMF behavior is observable compared to the 500×300 km grid depicted in Figure 4. Across all members of the satellite constellation—Alpha, Beta, and Charlie—more pronounced waveform behavior shifts are evident leading up to and following the seismic event. This divergence arises from the expanded coverage area of our geospatial boundary, resulting in a greater number of data samples.

Figure 7a shows the results obtained using the original Martingale system with the data plotted in Figure 6 for each of the satellite time-series. Starting with Alpha, we see a number of anomalies detected up to 16 months prior to the seismic event data until roughly 5 months prior, followed by a period of no detection until 1 month post the event date; it is noteworthy that Alpha exhibits a series of large spikes causing anomaly detection, which are sustained for long periods of time. Beta begins to exhibit triggering behavior at approximately 9 months prior to the event data, with strong spiking activity over a long duration of time until 3 months prior to the event data; no further anomalies are detected afterwards. Charlie only exhibits anomalies approximately 1 year prior to the event data and remains untriggered, with no anomalies detected for the remainder of the time-series.

We do note that, although no anomalies are detected near the event data, the time-series themselves show some spiking activity with long durations just before and after the event data for all three satellites when processed with *z*-Martingale.

Figure 7b depicts the outcomes derived from employing our *z*-Martingale system with the data plotted in Figure 6 for each of the satellite time-series. In comparison to the original Martingale system (Figure 7a), we observe fewer anomalies for Alpha, with none detected approximately 1 year before the seismic event. Regarding Beta, the results closely resemble those obtained in Figure 7a, albeit with the *z*-Martingale system producing fewer anomalies but larger spikes in magnitude. Similarly, anomalies for Charlie appear similar to those in Figure 7a, although the spikes exhibit greater power in terms of magnitude. Additionally, the time-series produces anomalies just after the event date.

We do note that, as before, the time-series themselves show some spiking activity with long durations just before and after the event data for all three satellites when processed with *z*-Martingale.

### 4.2. Greece 2020

On 30 October 2020, a seismic event with a moment magnitude of 7.0 occurred approximately 14 km northeast of the Greek island of Samos. While Samos was closest to the epicenter, the Turkish city of İzmir, located 70 km northeast, bore the brunt of the impact. Over 700 residential and commercial structures in İzmir were seriously damaged or destroyed as a result. This earthquake, though not unprecedented in the region, had significant consequences due to its intensity and proximity to populated areas. The seismic activity served as a reminder of the vulnerability of communities situated along active fault lines and highlighted the importance of preparedness and resilience in earthquake-prone regions. In the aftermath, relief efforts focused on providing aid to affected areas and implementing measures to mitigate the impact of future seismic events. The event underscored the need for continued research and investment in earthquake monitoring and early warning systems to enhance disaster response and minimize loss of life and property in seismic zones.

Greece is located in a seismically active region characterized by frequent earthquakes. The country is situated at the convergence of several tectonic plates, including the Eurasian Plate, the African Plate, and the Anatolian Plate. These plates interact along the Hellenic Arc, a major geologic feature that extends through Greece and neighboring countries. As a result of this tectonic activity, Greece experiences a relatively high frequency of earthquakes of varying magnitudes. Some regions in Greece, particularly those near fault lines and plate boundaries, are considered to be at higher risk of seismic activity compared to others. This high activity is evident in both our 500×300 km and 1000×800 km plots; Figure 8a,b and Figure 9a,b.

#### 4.2.1. 500×300 km

Figure 10 showcases the EMF intensity activity across a 500×300 km area before and after the seismic event. Each data point represents the mean intensity value per day, calculated using Equation (Equation 10). The figure consists of two sub-figures: Figure 10a,b.

In Figure 10a, the unmodified time-series obtained from the 500×300 km grid is depicted. It is noticeable that approximately 1 year prior to the event date, the values obtained for Beta exhibit errors, showing zero values for $b_{x}$, $b_{y}$, and $b_{z}$. To enhance visual clarity, Figure 10b displays only the time-series of Alpha and Charlie. It is important to highlight that although we modified the plot in Figure 10b to exclude Beta, we did not exclude Beta from our analysis. We believe both the original Martingale and *z*-Martingale systems to be robust enough to handle such errors appropriately.

Figure 8 shows the results from both the Martingale and *z*-Martingale systems operating over the data shown in Figure 10a. Figure 8a shows the results obtained from the original Martingale system, with only Beta and Charlie producing anomalies. Beta produces roughly five anomaly detections just prior to the event date, while Charlie produces a number of anomalies through its time-series from roughly 10 months prior to 2 months post the seismic event with a large spike occurring near the same position as the anomalies detected by Charlie. Interestingly, we can observe an unusual waveform in Beta at the rough location of the erroneous data, the impact of which was minimized by the Martingale process but suggests the potential exploration of shape-based encoding [19] for future work.

Figure 8b shows the results of our *z*-Martingale, with all three satellites (Alpha, Beta, and Charlie) exhibiting anomalies in their time-series. The Alpha time-series produces anomalies at roughly 6 and 3 months prior to the seismic event; however, we note rapid fluctuations in EMF intensity lasting for long periods of time. Beta produces anomalies approximately 3 months post the event data, which is in contrast to Figure 8a where Beta produces anomalies just before the event date; again, as in Figure 8a, we can observe an unusual waveform pattern for Beta at the time of the erroneous data sampling. The time series of Charlie appears to follow a similar pattern to Charlie of Figure 8a but with fewer anomalies.

#### 4.2.2. 1000×800 km

Figure 11 showcases the EMF intensity activity across a 1000×800 km area before and after the seismic event. Each data point represents the mean intensity value per day, calculated using Equation (Equation 10). The figure consists of two sub-figures (as in Section 4.2.1): Figure 11a,b.

In Figure 11a, the unmodified time-series obtained from the 1000×800 km grid is depicted. It is noticeable that approximately 1 year prior to the event date, the values obtained for Beta exhibit errors, showing zero values for $b_{x}$, $b_{y}$, and $b_{z}$. To enhance visual clarity, Figure 11b displays only the time-series of Alpha and Charlie. It is important to highlight that, as in Section 4.2.1, though we modified the plot in Figure 11b to exclude Beta, we did not exclude Beta from our analysis.

Figure 9 shows the results from both the Martingale and *z*-Martingale systems operating over the data shown in Figure 11a. Figure 9a shows the results obtained from the original Martingale system. Alpha, Beta, and Charlie allow the production of anomalies within their respective time-series. Alpha exhibits a group of anomalies approximately 10 months prior to the seismic event date followed by anomalies immediately preceding and on the event date. Beta exhibits anomalies at approximately 9 months prior to the event date with no further detection afterwards. Charlie’s time series results in several anomalies between the 9 month and 6 month periods of time without any further reports.

Figure 9b shows the results obtained from processing the data of Alpha, Beta, and Charlie with our *z*-Martingale approach. Alpha follows a similar pattern of detected anomalies as in Figure 9a, though with larger visible spikes. Beta exhibits a similar behavior as in Figure 9a, though more anomalies are produced and several spikes show significant increases in their magnitudes. Charlie follows the pattern exhibited by Figure 9a very closely, with anomalies detected at approximately the same time; interestingly, as with Alpha and Beta, we see distinctive change in the magnitude of spikes within Charlie. Across all of the signals, it is noteworthy that the waveform through the experiment is different when comparing Martingale and *z*-Martingale, indicating the potential for future research utilizing waveform analytics and pattern recognition techniques.

### 4.3. Croatia 2020

On 29 December 2020, Croatia experienced a seismic event with significant implications for the region. The earthquake, with a magnitude of 6.4 and located within central Croatia, underscored the country’s susceptibility to seismic activity despite not being as seismically active as other Mediterranean regions. While the event’s magnitude may not have been unprecedented, its impact on local communities and infrastructure was notable. Croatia, situated in southeastern Europe, experiences seismic activity due to its location at the boundary of the Eurasian Plate and the Adriatic microplate. While not as seismically active as some other Mediterranean regions, Croatia still faces occasional earthquakes resulting from the collision and subduction of these tectonic plates, along with the presence of fault systems within the region. The seismicity in Croatia, though generally of lower magnitude, poses risks to local communities and infrastructure. In this case study, we delve into a specific earthquake event that occurred in Croatia on a particular date and location. Our focus lies on analyzing the electromagnetic field (EMF) intensity data collected before and after this seismic event.

Like our study in Section 4.2, the area of investigation time-series extracted for Beta contains a zero error, with the values for $b_{x}$, $b_{y}$, and $b_{z}$ equaling zero. This is a common occurrence in long-form satellite telemetry and sensing.

#### 4.3.1. 500×300 km

Figure 12 showcases the EMF intensity activity across a 500×300 km area before and after the seismic event. Each data point represents the mean intensity value per day, calculated using Equation (Equation 10). The figure consists of two sub-figures (as in Section 4.2.1): Figure 13a,b. In Figure 13a, the unmodified time-series obtained from the 1000×800 km grid is depicted. It is noticeable that approximately 1 year prior to the event date, the values obtained for Beta exhibit errors, showing zero values for $b_{x}$, $b_{y}$, and $b_{z}$. To enhance visual clarity, Figure 13b displays only the time-series of Alpha and Charlie. It is important to highlight that, as in Section 4.2.1, though we modified the plot in Figure 13b to exclude Beta, we did not exclude Beta from our analysis.

Figure 12 shows the plots obtained from processing the data shown in Figure 13a using the Martingale and *z*-Martingale systems for Alpha, Beta, and Charlie EMF time-series produced over a 500×300 km grid centered on the epicenter of the 2020 Croatian earthquake. Figure 12a shows the results of using the Martingale system to detect anomalies on the EMF data. It is observable that no anomalies are detected by the Martingale through the entire time-series for Alpha; the Beta time-series produces anomalies at roughly 1 month post the event date; and Charlie produces some anomalies roughly 4 months prior to the event date. It is noteworthy to highlight the waveform variations that occur through the plotting.

Figure 12b shows the anomalies detected using the *z*-Martingale system. Alpha produces some anomalies ahead of the event date, in contrast with the plot in Figure 12a; notably, anomalies are detected at roughly 1 year and 4 months prior to the event date. Beta produces anomalies at 1 year and 5 months prior to the event date and 1 month following. Charlie produces anomalies at roughly 1 year and four months.

#### 4.3.2. 1000×800 km

Figure 14 showcases the EMF intensity activity across a 1000×800 km area before and after the seismic event. Each data point represents the mean intensity value per day, calculated using Equation (Equation 10). The figure consists of two sub-figures (as in Section 4.3.1): Figure 14a,b. In Figure 14a, the unmodified time-series obtained from the 1000×800 km grid is depicted. It is noticeable that approximately 1 year prior to the event date, the values obtained for Beta exhibit errors, showing zero values for $b_{x}$, $b_{y}$, and $b_{z}$. To enhance visual clarity, Figure 14b displays only the time-series of Alpha and Charlie. It is worth emphasizing that, similar to Section 4.3.1, the plot in Figure 14b was adjusted to exclude Beta. However, it is crucial to note that Beta was not excluded from our analysis.

Figure 15 shows the results from both the Martingale and *z*-Martingale systems operating over the Swarm constellation intensity data shown in Figure 14. Figure 15a shows the results obtained from the Martingale system operating over the data. Figure 15b shows the results obtained from operating with the *z*-Martingale system. For Figure 15a, taking each satellite in turn, we observe the following: the Alpha time-series contains anomalies detected at 1 year, 6 months, 3 months, and 2 months prior to the event date with no further alarms; the Beta time-series contains anomalies detected at roughly 3 months post the event date with no previous anomalies detected; the Charlie time-series contains anomalies detected at over 1 year, 1 year, 8 months, and 2 months prior to the event date as well as roughly 1 month following the event date. Consider the *z*-Martingale of Figure 15b where we observe the following satellite time-series behaviors: the Alpha time-series contains anomalies at 6 months, 3 months, and 2 months prior to the event (similar to the observable anomalies in Figure 15a; the Beta time-series contains anomalies detected at roughly 3 months post the event date with no previous anomalies detected; the Charlie time-series contains many anomalies detected at over 1 year, 1 year, 8 months, and 2 months prior to the event data as well as roughly 1 month following the event date.

## 5. Discussion

The quantitative experiments reveal a notable performance enhancement of our novel approach over the original Martingale framework, especially when considering the r-rank metric. This signifies that our method effectively identifies anomalies of higher importance, based on their rank, without the necessity of pre-defining a threshold. This flexibility stands as a significant advantage over the original framework. Moreover, these results instill confidence in our methodology, demonstrating its ability to enhance sensitivity to complex temporal relationships without relying on a dynamic threshold.

In our qualitative assessment, we apply both our novel approach and the original Martingale framework to real-world earthquake case studies in Mexico, Greece, and Croatia. Overall, our framework tends to detect more anomalies across all case studies compared to the original framework, particularly when considering data from the Beta satellites, which were excluded from the graphical representation, on which our method does not perform as well.

However, both methods show inadequacies when compared against manually identified anomalies from ground truth data as evident with Table 3, though the removal of the fixed thresholding is a clear advantage toward *z*-Martingale. This raises questions about the reliability and robustness of the Martingale and *z*-Martingale methods, suggesting a need for substantial improvement or a shift toward more dependable methodologies. It is interesting that the *z*-Martingale is visibly more robust in the detection of anomalies as well as emphasizing/retaining waveform sensitivity.

Despite the quantitative evidence indicating the capability of our approach in identifying stronger anomalies, it raises concerns when considered alongside the claim that our method is adept at detecting subtle changes. The discrepancy between detecting subtle anomalies statistically while missing obvious ones prompts reflection on the reliability of the hypothesis test method. Furthermore, the ambiguity surrounding the interpretation of these statistical anomalies as false positives underscores the need for a deeper understanding of the physical implications of the detected anomalies.

## 6. Conclusions

This paper presents a novel probabilistic model based on Martingale theory, which represents an advancement over previous methodologies by leveraging the *z*-space to eliminate the need for a predetermined threshold. Our primary objective is to detect anomalies in EMF activity, potentially serving as precursors to seismic events. To comprehensively assess our framework, we conducted two distinct experiments: a quantitative study facilitating rigorous statistical performance analysis and a qualitative investigation centered around three prominent earthquake case studies. The results of our experiments reveal a discernible improvement in performance compared to traditional approaches, underscoring our framework’s effectiveness in detecting early signs of abnormal EMF behavior preceding seismic events.

Throughout our inquiry, we identified several noteworthy concerns and promising avenues for future research. These include delving into the intricacies of electromagnetic patterns to enhance the accuracy of seismic anomaly detection; scrutinizing the reliability of the hypothesis test method; delving deeper into the nature of false positive results to refine anomaly detection algorithms; investigating the disparate sensitivities of different approaches to anomalies across various segments of time-series data; grid size analysis; and integrating anomaly graphs with ground truth data to facilitate transparent analysis, waveform analysis, shape encoding, and anomaly pattern recognition. Furthermore, we propose extending our research endeavors by integrating advanced deep learning techniques and further refining probabilistic models to scrutinize seismic event data for nuanced abnormalities, thereby enhancing the robustness of our framework in seismic precursor detection scenarios.

## Figures and Tables

**Figure 1 sensors-24-03654-f001:**
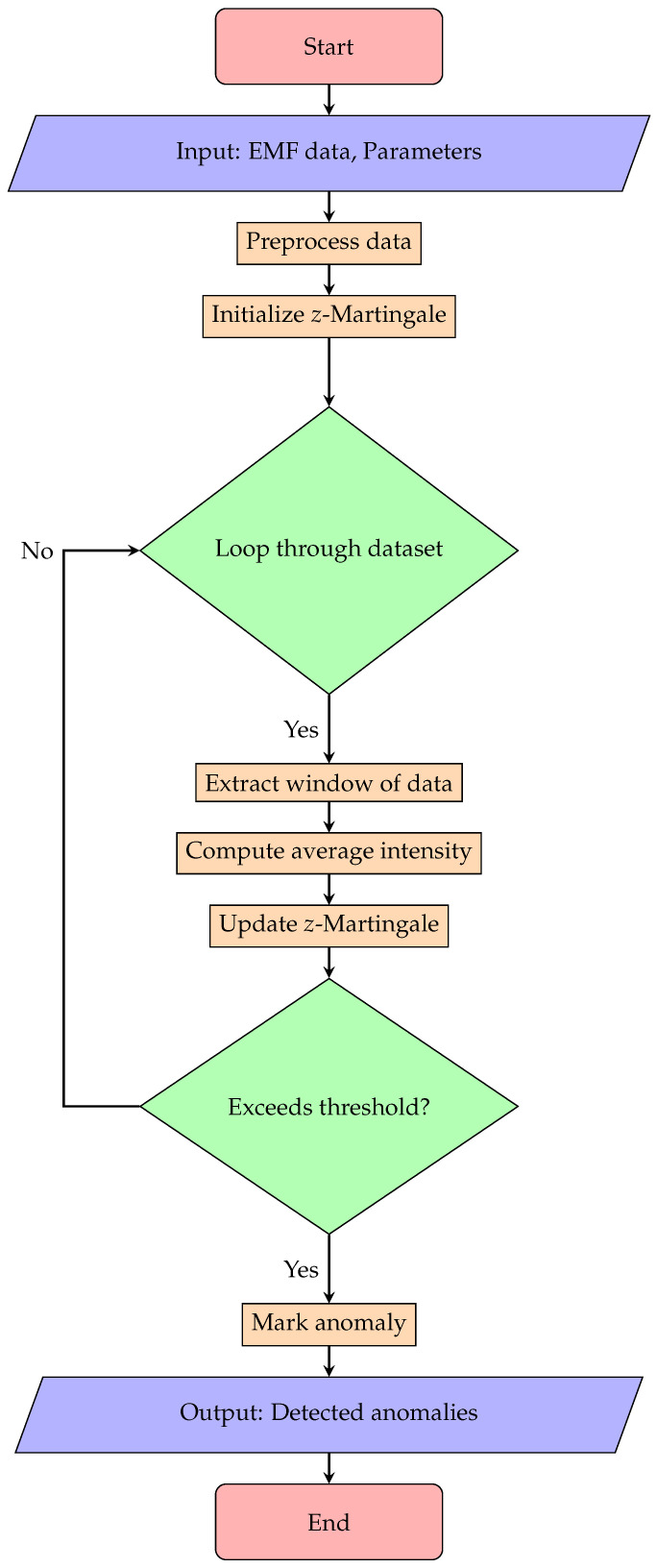
A flowchart illustrating our *z*-Martingale algorithm for anomaly detection in EMF signals.

**Figure 2 sensors-24-03654-f002:**
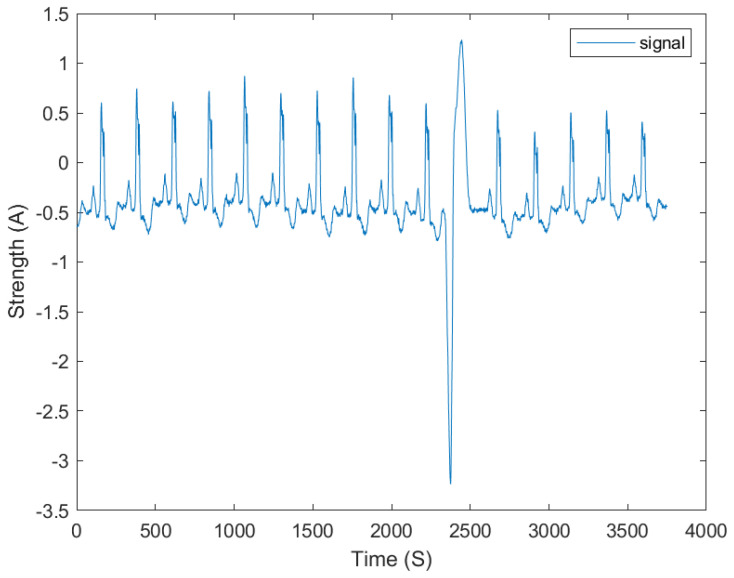
A sample of an ECG plot produced from the BIDMC Congestive Heart Failure Database data used in this project.

**Figure 3 sensors-24-03654-f003:**
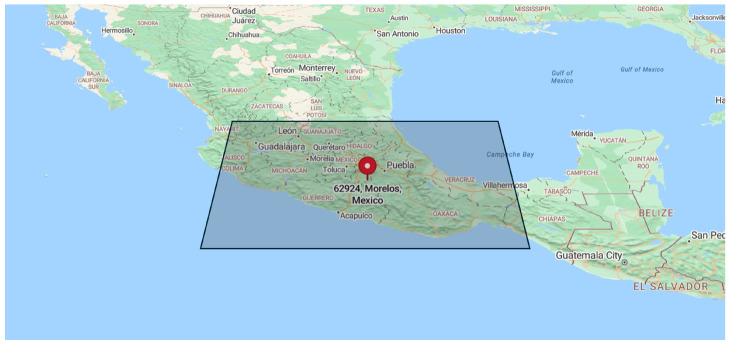
A tilted plot of our 500×300km area of interest with the epicenter of the Mexico 2017 earthquake indicated with a pin.

**Figure 4 sensors-24-03654-f004:**
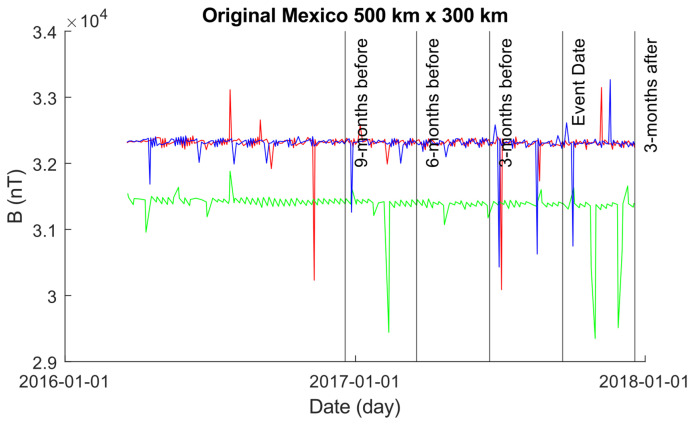
A plot of the original data occurring within the 500×300 km grid obtained using the original Swarm alongside Equation (Equation 10). The color coding is Alpha (red), Beta (green), and Charlie (blue).

**Figure 5 sensors-24-03654-f005:**
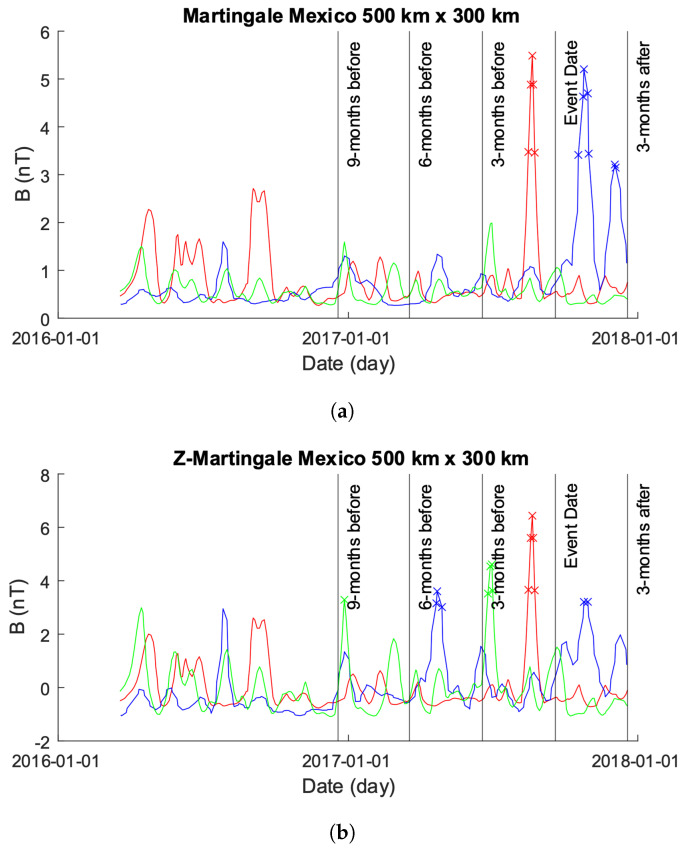
Comparison of Martingale and *z*-Martingale values within the 500×300 km grid for the Mexico case study. The color coding is Alpha (red), Beta (green), and Charlie (blue). (**a**) Martingale value data plotted within the 500×300 km grid obtained using the original Swarm satellite data alongside Equation (Equation 3). It is observable that this approach detects fewer anomalies compared to the *z*-Martingale (**b**) method. (**b**) *z*-Martingale value data plotted within the 500×300 km grid obtained using the original Swarm satellite data alongside Equation (Equation 7). It is observable that the *z*-Martingale scores produce more anomaly detections than the regular Martingale method (**a**) method.

**Figure 6 sensors-24-03654-f006:**
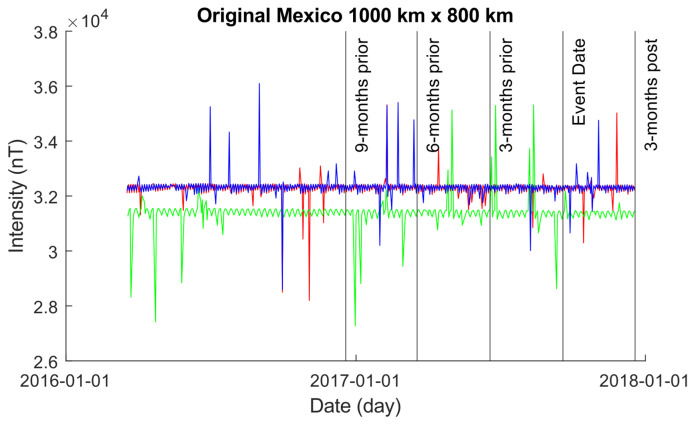
A plot of the original data occurring within the 1000×800 km grid obtained using the original Swarm alongside Equation (Equation 10). The color coding is Alpha (red), Beta (green), and Charlie (blue).

**Figure 7 sensors-24-03654-f007:**
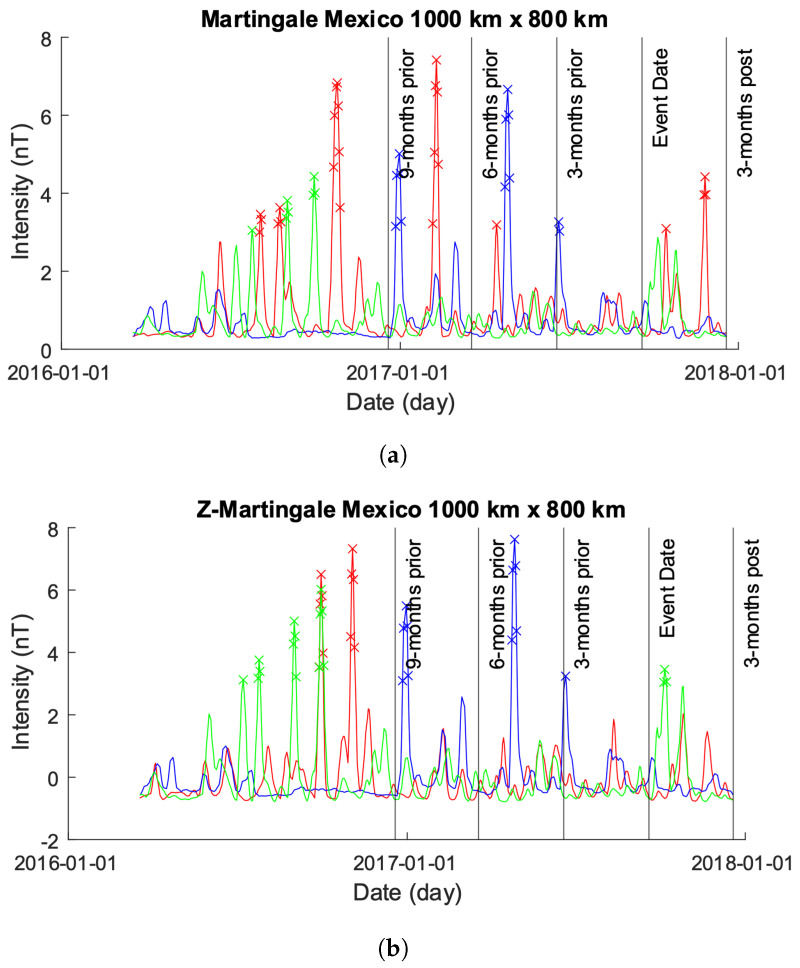
Comparison of Martingale and *z*-Martingale values within the 1000×800 km grid for the Mexico case study. The color coding is Alpha (red), Beta (green), and Charlie (blue). (**a**) Martingale value data plotted within the 1000×800 km grid obtained using the original Swarm satellite data alongside Equation (Equation 3). It is observable that this approach detects fewer anomalies compared to the *z*-Martingale (**b**) method. (**b**) *z*-Martingale value data plotted within the 1000×800 km grid obtained using the original Swarm satellite data alongside Equation (Equation 7). It is observable that the *z*-Martingale scores produce more anomaly detections than the regular Martingale method (**a**) method.

**Figure 8 sensors-24-03654-f008:**
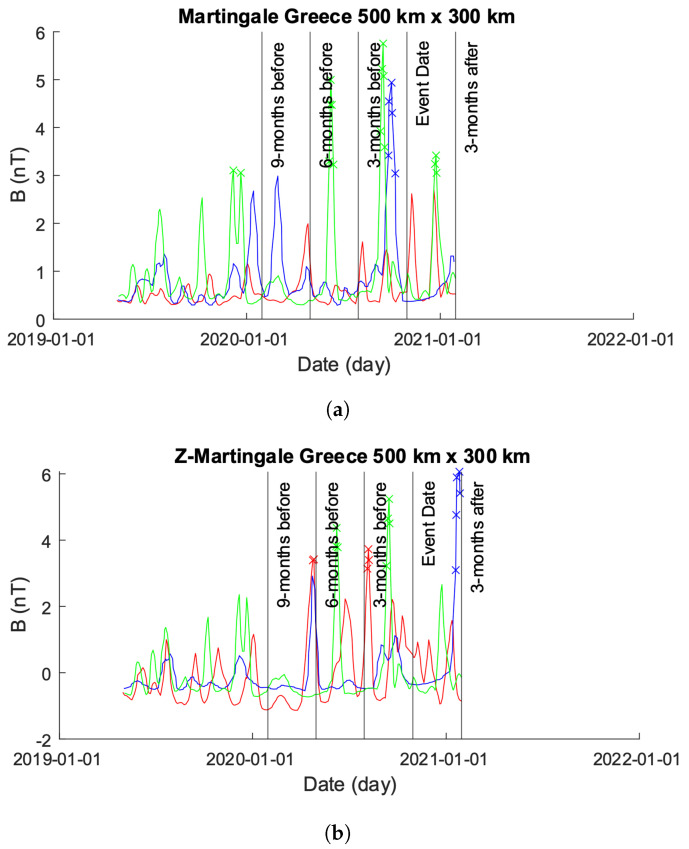
Comparison of Martingale and *z*-Martingale values within the 500×300 km grid for the Greece case study. The color coding is Alpha (red), Beta (green), and Charlie (blue). (**a**) Martingale value data plotted within the 500×300 km grid obtained using the original Swarm satellite data alongside Equation (Equation 3). It is observable that this approach detects fewer anomalies compared to the *z*-Martingale (**b**) method. (**b**) *z*-Martingale value data plotted within the 500×300 km grid obtained using the original Swarm satellite data alongside Equation (Equation 7). It is observable that the *z*-Martingale scores produce more anomaly detections than the regular Martingale method (**a**) method.

**Figure 9 sensors-24-03654-f009:**
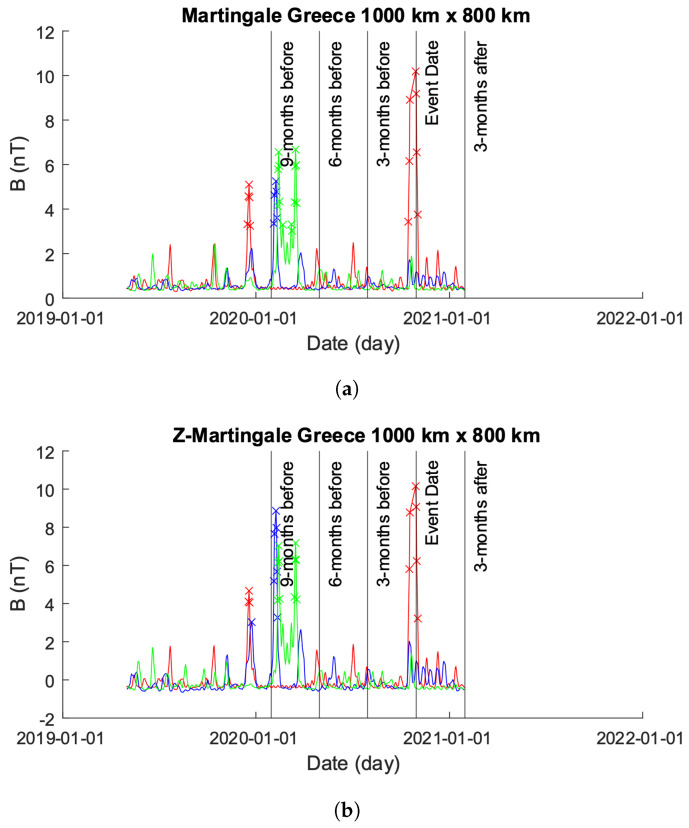
Comparison of Martingale and *z*-Martingale values within the 1000×800 km grid for the Greece case study. The color coding is Alpha (red), Beta (green), and Charlie (blue). (**a**) Martingale value data plotted within the 1000×800 km grid obtained using the original Swarm satellite data alongside Equation (Equation 3). It is observable that this approach detects fewer anomalies compared to the *z*-Martingale (**b**) method. (**b**) *z*-Martingale value data plotted within the 1000×800 km grid obtained using the original Swarm satellite data alongside Equation (Equation 7). It is observable that the *z*-Martingale scores produce more anomaly detections than the regular Martingale method (**a**) method.

**Figure 10 sensors-24-03654-f010:**
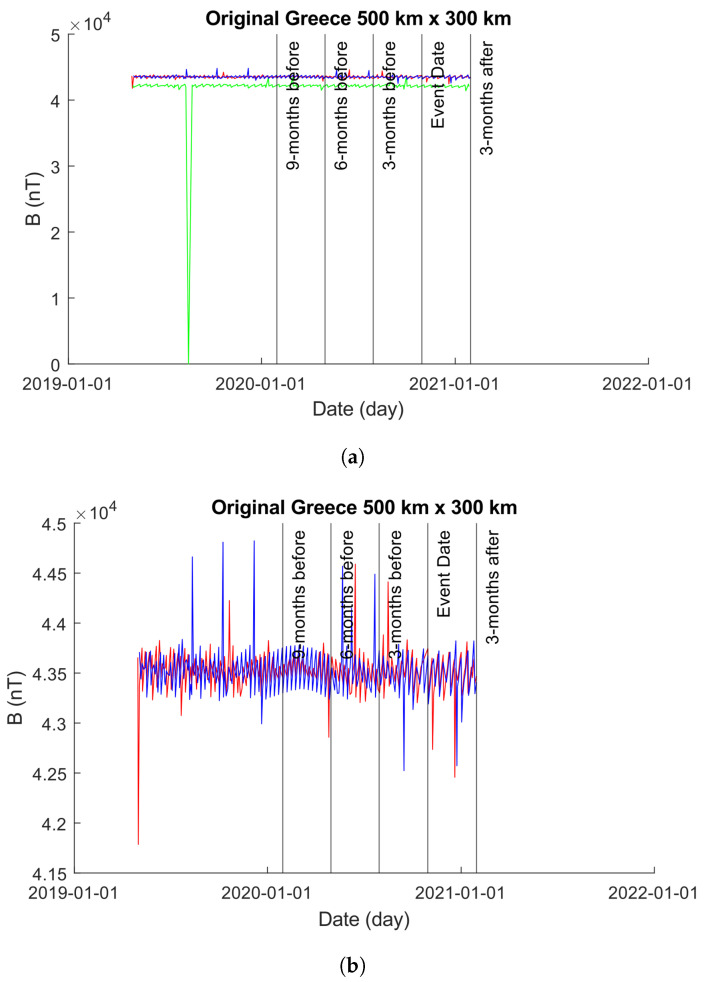
Plots of the time-series produced from Alpha, Beta, and Charlie satellites with intensity values obtained from a 500×300 km grid centered on the epicenter of the 2020 Greek earthquake. The color coding is Alpha (red), Beta (green), and Charlie (blue). (**a**) An unmodified plot of the time-series produced from Alpha, Beta, and Charlie over a 500×300 km grid centered on the epicenter of the 2020 Greek Earthquake. It is clear that the results of Beta inhibit our ability to visually explore the plot with ease, so we provide (**b**) with Beta removed to highlight the behaviors of Alpha and Charlie. (**b**) A modified plot of the time-series produced from Alpha and Charlie over a 500×300 km grid centered on the epicenter of the 2020 Greek Earthquake. Beta has been removed from the plot to enhance our visual interpretations but is observable in (**b**).

**Figure 11 sensors-24-03654-f011:**
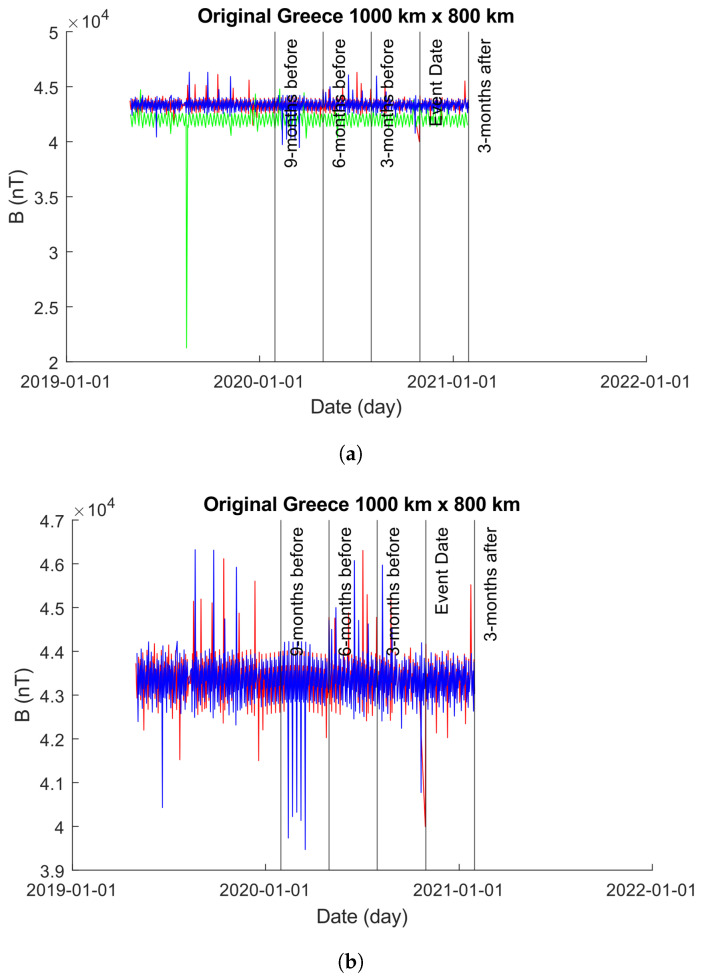
Plots of the time-series produced from Alpha, Beta, and Charlie satellites with intensity values obtained from a 1000×800 km grid centered on the epicenter of the 2020 Greek earthquake. The color coding is Alpha (red), Beta (green), and Charlie (blue). (**a**) An unmodified plot of the time-series produced from Alpha, Beta, and Charlie over a 1000×800 km grid centered on the epicenter of the 2020 Greek Earthquake. It is clear that the results of Beta inhibit our ability to visually explore the plot with ease, so we provide (**b**) with Beta removed to highlight the behaviors of Alpha and Charlie. (**b**) A modified plot of the time-series produced from Alpha and Charlie over a 1000×800 km grid centered on the epicenter of the 2020 Greek Earthquake. Beta has been removed from the plot to enhance our visual interpretations but is observable in (**a**).

**Figure 12 sensors-24-03654-f012:**
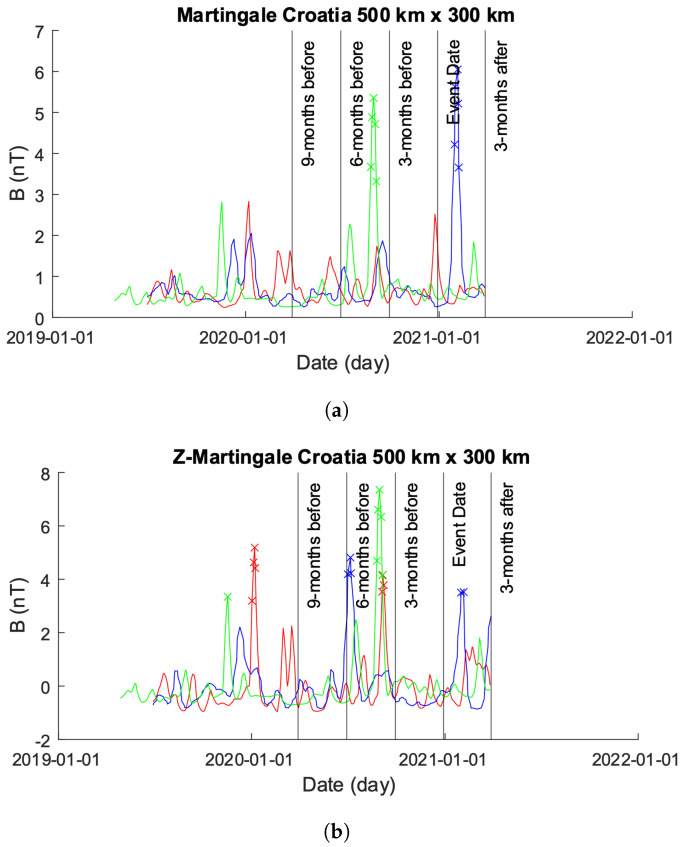
Comparison of Martingale and *z*-Martingale values within the 500×300 km grid for the Croatia case study. The color coding is Alpha (red), Beta (blue), and Charlie (green). (**a**) Martingale value data plotted within the 500×300 km grid obtained using the original Swarm satellite data alongside Equation (Equation 3). It is observable that this approach detects fewer anomalies compared to the *z*-Martingale (**b**) method. (**b**) *z*-Martingale value data plotted within the 500×300 km grid obtained using the original Swarm satellite data alongside Equation (Equation 7). It is observable that the *z*-Martingale scores produce more anomaly detections than the regular Martingale method (**a**) method.

**Figure 13 sensors-24-03654-f013:**
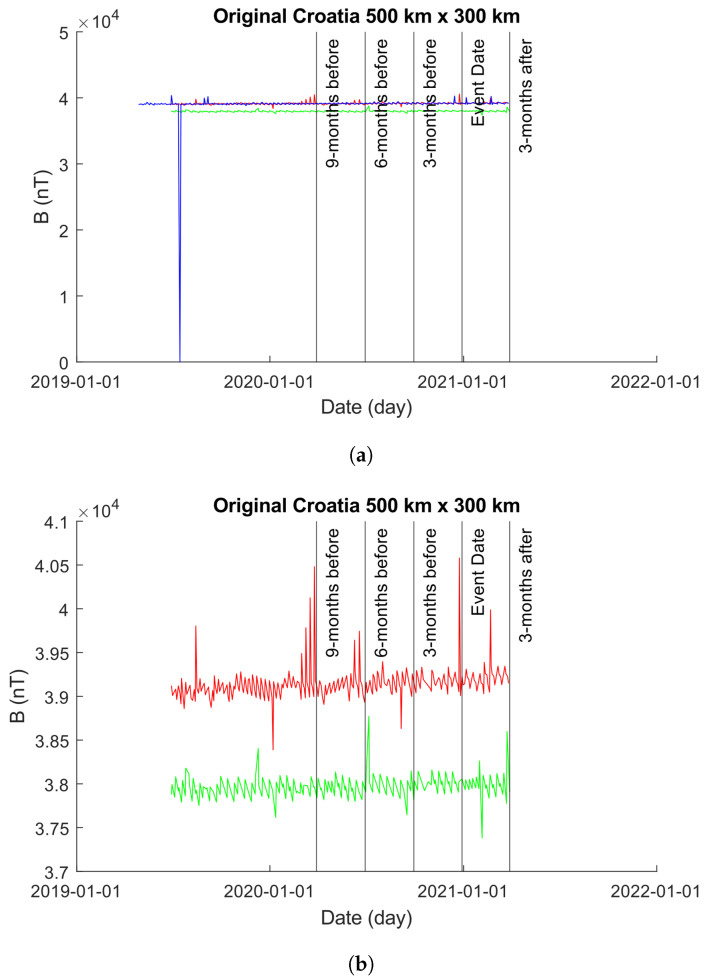
Plots of the time-series produced from Alpha, Beta, and Charlie satellites with intensity values obtained from a 500×300 km grid centered on the epicenter of the 2020 Croatia earthquake. The color coding is Alpha (red), Beta (blue), and Charlie (green). (**a**) An unmodified plot of the time-series produced from Alpha, Beta, and Charlie over a 500×300 km grid centered on the epicenter of the 2020 Croatian Earthquake. It is clear that the results of Beta inhibit our ability to visually explore the plot with ease, so we provide (**b**) with Beta removed to highlight the behaviors of Alpha and Charlie. (**b**) A modified plot of the time-series produced from Alpha and Charlie over a 500×300 km grid centered on the epicenter of the 2020 Croatian Earthquake. Beta has been removed from the plot to enhance our visual interpretations but is observable in (**a**).

**Figure 14 sensors-24-03654-f014:**
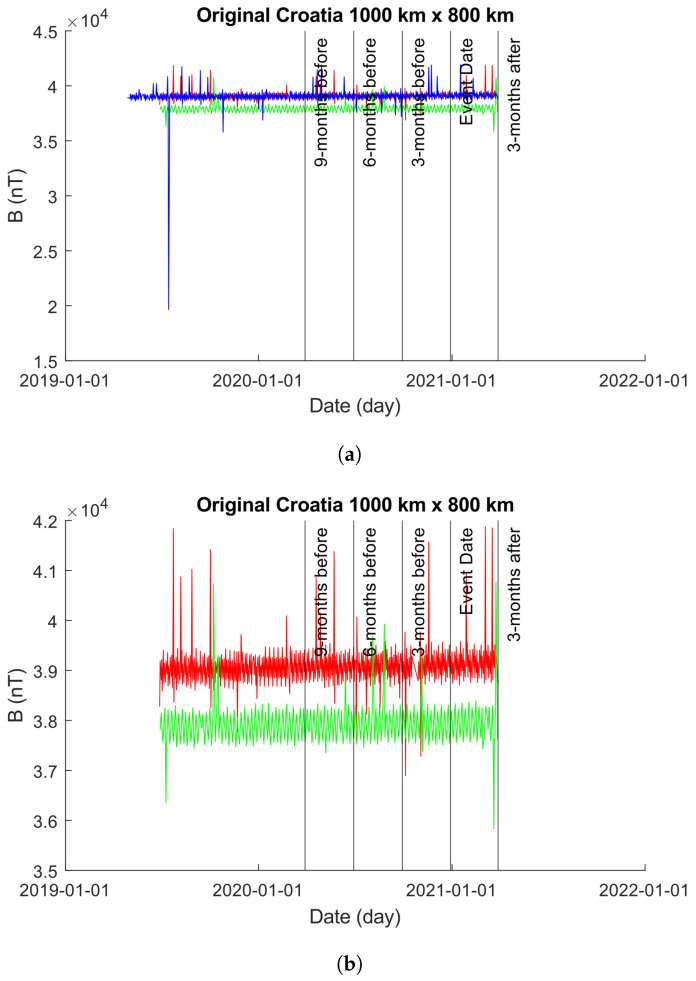
Plots of the time-series produced from Alpha, Beta, and Charlie satellites with intensity values obtained from a 1000×800 km grid centered on the epicenter of the 2020 Croatia earthquake. The color coding is Alpha (red), Beta (blue), and Charlie (green). (**a**) An unmodified plot of the time-series produced from Alpha, Beta, and Charlie over a 1000×800 km grid centered on the epicenter of the 2020 Croatian Earthquake. It is clear that the results of Beta inhibit our ability to visually explore the plot with ease, so we provide (**b**) with Beta removed to highlight the behaviors of Alpha and Charlie. (**b**) A modified plot of the time-series produced from Alpha and Charlie over a 1000×800 km grid centered on the epicenter of the 2020 Croatian Earthquake. Beta has been removed from the plot to enhance our visual interpretations but is observable in (**a**).

**Figure 15 sensors-24-03654-f015:**
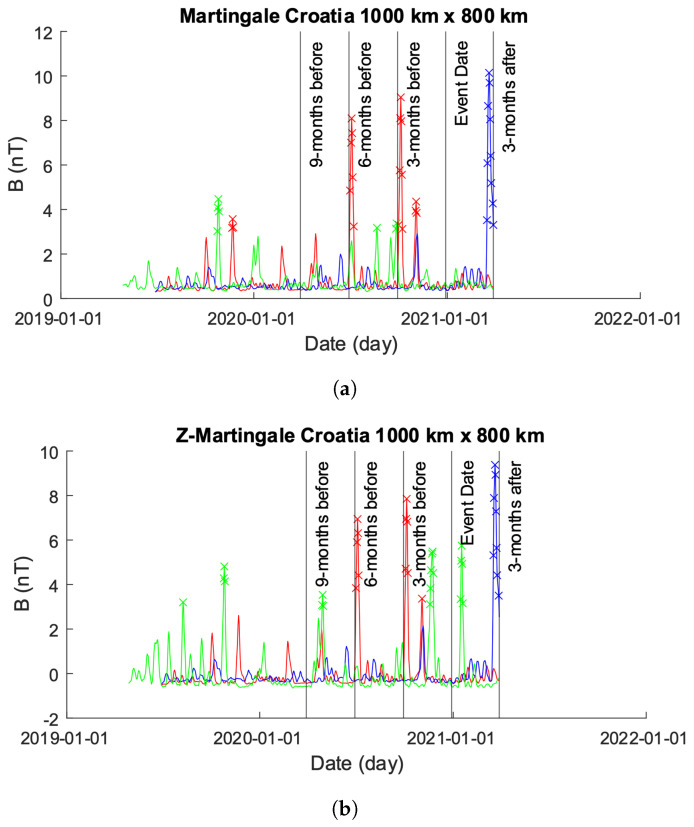
Comparison of Martingale and *z*-Martingale values within the 1000×800 km grid for the Greece case study. The color coding is Alpha (red), Beta (blue), and Charlie (green). (**a**) Martingale value data plotted within the 1000×800 km grid obtained using the original Swarm satellite data alongside Equation (Equation 3). It is observable that this approach detects fewer anomalies compared to the *z*-Martingale (**b**) method. (**b**) *z*-Martingale value data plotted within the 1000×800 km grid obtained using the original Swarm satellite data alongside Equation (Equation 7). It is observable that the *z*-Martingale scores produce more anomaly detections than the regular Martingale method (**a**).

**Table 1 sensors-24-03654-t001:** Results of original Martingale.

Dataset	Accuracy (%)	Precision (%)	R-Rank (%)
Chfdb_chf01_275_f1	0.91	0.79	0.26
Chfdb_chf01_275_f2	0.87	0.74	0.80
Chfdb_chf13_45590_f1	0.91	0.55	0.26
Chfdb_chf13_45590_f2	0.94	0.75	0.28
Data4	0.96	0.93	0.17

**Table 2 sensors-24-03654-t002:** Results of *Z*-score enhanced Martingale.

Dataset	Accuracy (%)	Precision (%)	R-Rank (%)
Chfdb_chf01_275_f1	0.91	0.75	0.62
Chfdb_chf01_275_f2	0.89	0.75	0.81
Chfdb_chf13_45590_f1	0.91	0.57	0.64
Chfdb_chf13_45590_f2	0.94	1.0	0.55
Data4	0.97	0.94	0.39

**Table 3 sensors-24-03654-t003:** Anomaly detection numbers for Martingale and *Z*-Martingale methods.

Case Study	Mexico (500 × 300)	Mexico (1000 × 800)	Greece (500 × 300)	Greece (1000 × 800)	Croatia (500 × 300)	Croatia (1000 × 800)
Martingale	3	5	21	21	17	20
*Z*-Martingale	4	5	10	9	6	10

## Data Availability

The findings presented in this study are underpinned by data accessible through the “VirES” platform, available at https://vires.services (accessed on 1 March 2024), with reference number [7]. These datasets, integral to our research, originate from the publicly accessible “Swarm-core” resource, accessible at https://earth.esa.int/eogateway/catalog/swarm-core (accessed on 1 March 2024). The availability of these datasets in the public domain exemplifies the spirit of open science and facilitates transparency and reproducibility in scientific research. Researchers and interested parties can access the data through the provided links, enabling further exploration and validation of the study outcomes.

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
