# Peer review of "Detection of Electromagnetic Seismic Precursors from Swarm Data by Enhanced Martingale Analytics"

_sensors, 2024, doi:10.3390/s24113654_

Round 1

Reviewer 1 Report

Comments and Suggestions for Authors

The paper "Detection of Electromagnetic Seismic Precursors from Swarm Data by An Enhanced Martingale Analytic" provide an updated analysis method based on martingales, exploiting data of satellite origin in the direction of electromagnetic precursors.

The work is interesting, but necessitates more clarity for what is the novelty and improvement to the martingale approach, as well as the type of data that are used and the

The Introduction needs to be improved, to let readers understand the context, what was done before, competitive methods and what are the needs that the present improvement addresses and satisfies. In addition there are some parts that are not clear at all:

- Lines 73-74. Original paper? which one? What is the "changepoint detection algorithm"? what it means and where it is described?

- References are not in order of citation: the first ones are [29] and [30].

- Line 94. Distance in space or in time? not clear.

Section 2 starts with "anomaly detection with Martingales", but it is not clarified which kind of anomaly. After it is understood that electromagnetic precursors are considered and this should be introduced, discussing briefly the physical standpoint to justify this type of precursors and if there is a justified chance of detection at some distance away. Thinking of satellites, you should clarify at which distance detection was carried out and if there are no false positives, so variations of the field that are incidental and not cause by the EQ.

Section 3. You should distinguish well what is applied of the "original Martingale" and what is the added for your proposed method.

Page 7, figure has no caption. The quantity in the y-axis is not clear.

Line 286 and following. Please, provide a source for these earthquakes where things like location, magnitude and depth are reported.
Table 3 should not be cited as blue and underlined.

Line 326: VFM?
To clarify which magnetic field (which frequency band) is used and if it is the one measured by the satellites or it is a reconstruction from those data at the soil level.

Line 454. The sentence suggests to include the Beta satellite data in future experiments. If I understand well, you could include this satellite now, show the results with and without, and provide your rationale including that. Have you done like this? If not, why?

Line 204, 612. The approach is said "novel": could you, please, synthesize where you have demonstrated this? (I cannot find it in the previous pages; I assume it should be where the martingales are described as at the beginning of section 3 it is already said "novel").

Uncompleted parts after the Conclusions related to autorship, materials, funding, etc.

References are not in MDPI style. They are also quite old with only 2 from the '20s one of which is a thesis dissertation.
There are no referenced papers from Sensors.

Comments on the Quality of English Language

Line 108 and elsewhere. Punctuation, such as a comma before "but".

Line 327 [REF]

Line 618: preformed

Author Response

Reviewer 1: The work is interesting, but necessitates more clarity for what is the novelty and improvement to the martingale approach, … The Introduction needs to be improved, to let readers understand the context, what was done before, competitive methods and what are the needs that the present improvement addresses and satisfies. In addition there are some parts that are not clear at all: Response: Thanks for the comments above, we have significantly reworked the manuscript to address the accessibility concern regarding the context.
- Lines 73-74. Original paper? which one? What is the "changepoint detection algorithm"? what it means and where it is described?
Response: The revision includes references to the original changepoint detection algorithm being enhanced here and rewrote accordingly.

- References are not in order of citation: the first ones are [29] and [30]. Response: This paper should now be in the correct format with references in proper order of appearance as required by Sensors.

Line 94. Distance in space or in time? not clear.
Response: Revised the sentence to “an effective distance over both space and time” as the parameters for space and time have both been varied and explored in the cited works combined.
Section 2 starts with "anomaly detection with Martingales", but it is not clarified which kind of anomaly. After it is understood that electromagnetic precursors are considered and this should be introduced, discussing briefly the physical standpoint to justify this type of precursors and if there is a justified chance of detection at some distance away. Thinking of satellites, you should clarify at which distance detection was carried out and if there are no false positives, so variations of the field that are incidental and not cause by the EQ.
Response: We have revised Section 2 to address this comment, including a change in title alongside more explanation of the problem area and our rationale of investigation.
Section 3. You should distinguish well what is applied of the "original Martingale" and what is the added for your proposed method.
Response: We have included further explanations and subsequent diagramming/context to better emphasis the difference between our proposed method and the original approach.
Page 7, figure has no caption. The quantity in the y-axis is not clear.

Response: This has been corrected.
-Line 286 and following. Please, provide a source for these earthquakes where things like location, magnitude and depth are reported.
Response: These have been added with context as footnotes on Page 9 as well as better in-text descriptions of the events.
Table 3 should not be cited as blue and underlined.

Response: This formatting issue has been resolved.
-Line 326: VFM? To clarify which magnetic field (which frequency band) is used and if it is the one measured by the satellites or it is a reconstruction from those data at the soil level.
Response: We clarify the use of the satellite data.
-Line 454. The sentence suggests to include the Beta satellite data in future experiments. If I understand well, you could include this satellite now, show the results with and without, and provide your rationale including that. Have you done like this? If not, why?

Response: To explain: Beta was only removed from the plotting for readability, we do not exclude it from analysis and repeat this emphasis in appropriate places.
-Line 204, 612. The approach is said "novel": could you, please, synthesize where you have demonstrated this? (I cannot find it in the previous pages; I assume it should be where the martingales are described as at the beginning of section 3 it is already said "novel").
Response: We have updated this to better illustrate the novelty as well as refer to appropriate sections with more appropriately placed in-depth details. Uncompleted parts after the Conclusions related to autorship, materials, funding, etc.
Response: We have completed these in this versioning.

References are not in MDPI style. They are also quite old with only 2 from the '20s one of which is a thesis dissertation.
Response: This has been adjusted in line with the comment.

There are no referenced papers from Sensors.

Response: We have included Sensor paper where appropriate which fit this paper’s work.

Reviewer 2 Report

Comments and Suggestions for Authors 1. The submitted manuscript presents an algorithm for detecting seismic activity precursors from the Earth’s electromagnetic field dataset. 2. The paper introduces a probabilistic model based on the Martingale probability theories to indicate abnormal changes in electromagnetic field activity and seems effective. 3. An enhancement of a changepoint detection algorithm utilising the martingale probability theory is added to the subject area. 4. The subject is interesting and suitable for the readers of sensors.  However the manuscript is not well-written.  Actually, it is hard to be read.  First, it is recommended to write a flowchart of the algorithm implementation  to help readers understand it.  Second, how to get r-rank in quantitative analysis? There is no explanation. It's not clear exactly what it means.   5. In the qualitative analysis, the relationship between the detection results of anomalies and the time of the earthquakes seems not obvious. For example, there are intensity spikes could be observed in Fig.6 both before and after the earthquake. The results of compared method in Fig. 7 show that the combined A and B satellites seem to see electromagnetic anomalies in the pre- and post-earthquake time area, whereas in the improved algorithm of Fig. 8 it seems that the anomalies appear over a long period of time of the earthquake, and the interrelationships are not clear enough. This needs to be clarified. 6. The references seems to be appropriate. 7.  The talbes and figures and formulas are so poorly formatted that it is difficult to evaluate the manuscript. The formula fonts are very bad, and a lot of the numbering is incorrect (e.g. those in pp. 7). There is no title for the figure in Page 7. We have no idea for what it means.  Comments on the Quality of English Language

Some expressions can be further condensed.

Author Response

1. The submitted manuscript presents an algorithm for detecting seismic activity precursors from the Earth’s electromagnetic field dataset.
2. The paper introduces a probabilistic model based on the Martingale probability theories to indicate abnormal changes in electromagnetic field activity and seems effective.
3. An enhancement of a changepoint detection algorithm utilising the martingale probability theory is added to the subject area.
4. The subject is interesting and suitable for the readers of sensors. However the manuscript is not well-written. Actually, it is hard to be read. First, it is recommended to write a flowchart of the algorithm implementation to help readers understand it. Second, how to get r-rank in quantitative analysis? There is no explanation. It's not clear exactly what it means.
Response: Thank you for your comments – we have designed and included a flowchart as suggested, placed it on Page 5. Additionally, we have included a further explanation of R-rank at Lines 214-218.
5. In the qualitative analysis, the relationship between the detection results of anomalies and the time of the earthquakes seems not obvious. For example, there are intensity spikes could be observed in Fig.6 both before and after the earthquake. The results of compared method in Fig. 7 show that the combined A and B satellites seem to see electromagnetic anomalies in the pre- and post-earthquake time area, whereas in the improved algorithm of Fig.
8 it seems that the anomalies appear over a long period of time of the earthquake, and the interrelationships are not clear enough. This needs to be clarified.
Response: We have revised the analysis section significantly to address these comments, including breaking the case studies up into their own sections and providing more results in-depth.
6. The references seems to be appropriate.
7. The tables and figures and formulas are so poorly formatted that it is difficult to evaluate the manuscript. The formula fonts are very bad, and a lot of the numbering is incorrect (e.g. those in pp. 7). There is no title for the figure in Page 7. We have no idea for what it means.
Response: We have revised the figures to increase their accessibility to general readers and have correct the number as noted.

Round 2

Reviewer 1 Report

Comments and Suggestions for Authors

No other remarks, except one point: as a reply to the last comment you said that you have included references from MDPI Sensors related to this work. I cannot see any.